# Of the Mechanisms of Paroxysmal Depolarization Shifts: Generation and Maintenance of Bicuculline-Induced Paroxysmal Activity in Rat Hippocampal Cell Cultures

**DOI:** 10.3390/ijms241310991

**Published:** 2023-07-01

**Authors:** Denis P. Laryushkin, Sergei A. Maiorov, Valery P. Zinchenko, Valentina N. Mal’tseva, Sergei G. Gaidin, Artem M. Kosenkov

**Affiliations:** Federal Research Center “Pushchino Scientific Center for Biological Research of the Russian Academy of Sciences”, Institute of Cell Biophysics of the Russian Academy of Sciences, 142290 Pushchino, Russia

**Keywords:** paroxysmal depolarization shift (PDS), Gi-coupled receptors, AMPA receptors, NMDA receptors, kainate receptors

## Abstract

Abnormal depolarization of neuronal membranes called paroxysmal depolarization shift (PDS) represents a cellular correlate of interictal spikes. The mechanisms underlying the generation of PDSs or PDS clusters remain obscure. This study aimed to investigate the role of ionotropic glutamate receptors (iGluRs) in the generation of PDS and dependence of the PDS pattern on neuronal membrane potential. We have shown that significant depolarization or hyperpolarization (by more than ±50 mV) of a single neuron does not change the number of individual PDSs in the cluster, indicating the involvement of an external stimulus in PDS induction. Based on this data, we have suggested reliable protocols for stimulating single PDS or PDS clusters. Furthermore, we have found that AMPA (α-amino-3-hydroxy-5-methyl-4-isoxazolepropionic acid) receptors are necessary for PDS generation since AMPAR antagonist NBQX completely suppresses bicuculline-induced paroxysmal activity. In turn, antagonists of NMDA (N-methyl-D-aspartate) and kainate receptors (D-AP5 and UBP310, respectively) caused a decrease in the amplitude of the first action potential in PDSs and in the amplitude of the oscillations of intracellular Ca^2+^ concentration occurring alongside the PDS cluster generation. The effects of the NMDAR (NMDA receptor) and KAR (kainate receptor) antagonists indicate that these receptors are involved only in the modulation of paroxysmal activity. We have also shown that agonists of some G_i_-coupled receptors, such as A_1_ adenosine (A_1_Rs) or cannabinoid receptors (CBRs) (N_6_-cyclohexyladenosine and WIN 55,212-2, respectively), completely suppressed PDS generation, while the A_1_R agonist even prevented it. We hypothesized that the dynamics of extracellular glutamate concentration govern paroxysmal activity. Fine-tuning of neuronal activity via action on G_i_-coupled receptors or iGluRs paves the way for the development of new approaches for epilepsy pharmacotherapy.

## 1. Introduction

Paroxysmal depolarization shift (PDS) is a prolonged (hundreds of milliseconds) depolarization of neuronal membrane potential by 20–70 mV occurring as a single event or as a cluster consisting of some PDSs [1]. PDS is assumed to be a neuronal correlate of the interictal spikes (IISs) detected with electroencephalography in the periods between epileptic discharges. Although the PDSs were first described in the 1970s, some important questions remain unanswered: (1) What is the role of PDSs in epileptogenesis? (2) Is PDS a manifestation of the intrinsic activity of a single neuron? (3) What mechanisms underlie the association of PDSs into clusters? (4) What factors determine the frequency and duration of individual PDSs and especially PDS clusters? Using rat hippocampal neuron-glial cell cultures, we have tried to address some of these questions in our present study.

The analysis of the literature data revealed that the role of PDSs/IISs in epilepsy is debatable since two opposite hypotheses have been formulated. According to the first one, IISs enhance the formation of abnormal connections, thus making the neuronal networks more excitable and promoting epileptogenesis [2,3]. According to the alternative hypothesis, IISs occur independently on ictal spikes and may decrease the inclination to seizures [4]. It may be possible that both hypotheses are true, but only for particular conditions and different time intervals. 

The generation of PDS accompanies IIS in a large number of neurons firing synchronously. Many distinct mechanisms, including communications via gap junctions, ephaptic interactions, and excitatory neurotransmission mediated by ionotropic glutamate receptors (iGluRs), such as NMDA and AMPA receptors (NMDARs and AMPARs, respectively), underlie the powerful synchronization of neurons [1,5]. It has been shown that antagonists of NMDARs and AMPARs partially or entirely suppressed the paroxysmal activity [6,7,8], whereas the role of kainate receptors (KARs) remains unknown. L-type voltage-gated calcium channels also significantly pattern PDSs. These channels determine the paroxysmal shift duration and the number of individual PDSs in a cluster [9,10]. Interestingly, the contribution of other voltage-gated channels has been poorly studied, which may be explained by the absence of a standard, reliable model of paroxysmal activity induction, including specific stimulation protocols. 

Different metabotropic receptors are considered the targets for the therapy of neurological diseases, including epilepsy. G-protein-coupled receptors are metabotropic receptors whose functions are determined by the type of associated G-protein. G-proteins consist of alpha (α) and beta-gamma subunits (βγ). They are divided into three main families based on the sequence similarity of the alpha subunits: G_s_-, G_q_-, and G_i_-proteins [11]. G_i_-coupled receptors are of great interest in terms of epilepsy treatment since their activation induces a decrease in neuronal activity. Members of this family present in all classes of receptors to main neuromodulators, including dopaminergic, serotoninergic, adenosine, and cannabinoid receptors [12]. Activation of G_i_-coupled receptors results in adenylyl cyclase inhibition followed by a decrease in intracellular cyclic adenosine monophosphate (cAMP) level. Interestingly, βγ- but not α subunit mediates the changes in ion channel activity, thus causing a decrease in intrinsic neuronal excitability and network activity due to attenuation of neurotransmitter release. 

In this study, we demonstrate the role of membrane potential and the contribution of iGluRs (especially kainate receptors) in the generation of single PDSs and PDS clusters in the model of bicuculline-induced paroxysmal activity, and discuss the mechanisms underlying the PDS induction. We also demonstrate the effects of agonists of different classes of G_i_-coupled receptors on paroxysmal activity.

## 2. Results

### 2.1. The Effects of Holding Potential on PDS Structure

Figure 1A demonstrates typical PDSs combined into a cluster. Each PDS in the cluster, excluding the first one, occurs before the restoration of membrane potential to the resting values. 

We denote membrane potential values at the moment of the next PDS generation in a cluster as PMP_min_. In this case, an individual PDS is characterized by abnormal membrane potential shift achieving −20 mV at the maximum (PMP_max_). Notably, PMP_max_ values are almost similar for each PDS in a cluster (Figure 1B), while the intervals between individual PDSs increase (Figure 1C). PDSs are accompanied by elevations of intracellular Ca^2+^ concentration ([Ca^2+^]_i_). Figure 1D shows that calcium oscillations (regular [Ca^2+^]_i_ elevations) occur in all neurons. These events are quasi-synchronous and delay accounts for approximately 50 ms for neurons in a viewfield. Moreover, high-frequency recordings demonstrate that oscillations include some peaks (Figure 1(D′)). Considering that in a network consisting of many neurons PDSs are combined, as a rule, into a cluster, it can be assumed that each peak in the calcium trace corresponds to a single PDS in the cluster. Based on these recordings, we have proposed that the number of PDSs in the cluster is similar for each neuron and accounts for three PDSs in the case of experiments shown in Figure 1(D′). Therefore, a paroxysmal shift detecting in one neuron almost synchronously occurs in other neurons in a network. Thus, bicuculline-induced paroxysmal activity is characterized by hyperexcitation and hypersynchronization of the neuronal network. 

We have performed the following experiments to understand better the structure of PDS and its effect on neurons. First, we gradually depolarized (Figure 2) and hyperpolarized (Figure 3) neurons during bicuculline-induced paroxysmal activity. 

Depolarization increased PMP_max_ (Figure 2D) and PMP_min_ (Figure 2E) values, but we observed regular PDS generation. Interestingly, the number of PDSs in a cluster did not change even at 0 mV (Figure 2A,C). Notably, most voltage-gated sodium and calcium channels are inactivated at this membrane potential, so action potentials cannot be generated at the beginning of the paroxysmal shift. The structure of the PDS is triangle-like in this case (Figure 2A, arrows 2–4) and is characterized by a fast potential rise followed by virtually uniform decay. In turn, PDS amplitude is similar during a cluster. These experiments demonstrate that each paroxysmal shift generated after the first one is caused by external stimuli occurring sequentially but not by the intrinsic activity of a neuron in response to a single powerful stimulus. Additionally, each input stimulus during the PDS cluster reveals similar amplitude and duration. 

Hyperpolarization also did not affect the number of PDSs in a cluster (Figure 3C), thus confirming the assumption about the extracellular origin of stimuli inducing a paroxysmal shift during the individual cluster. If the membrane potential is held at −100 mV, the neuron cannot achieve the threshold of action potential (AP) generation. Membrane hyperpolarization decreases PMP_max_ (Figure 3D) and PMP_min_ (Figure 3E) values, while the pattern of the response to paroxysmal depolarization also changes (Figure 3A). One AP is observed in PDSs generated at normal membrane potential (Figure 3A, arrows 1 and 2). In turn, hyperpolarization of membrane potential increased the number of APs (Figure 3A, arrows 3 and 4).

### 2.2. The Response of Neurons to Paroxysmal Depolarization

We have divided the neurons into two groups depending on their reaction to paroxysmal depolarization. Neurons from the first group (Figure 4A) respond to a paroxysmal shift by a single AP occurring at the moment of AP threshold overcome. In contrast, two or more APs follow the PDS in neurons of the second group (Figure 4B). Neurons from the first group demonstrate low-frequency firing in response to current injection (Figure 4(A′)). Due to their electrophysiological properties, these neurons cannot generate some APs during a PDS. In turn, fast-spiking neurons generating several consecutive APs (Figure 4(B′)) during the first 400 ms of stimulation belong to the second group. However, it is impossible to predict the response of particular neurons to a paroxysmal shift based only on the electrophysiological features since the pattern of spiking activity also depends on the stimulation intensity. Figure 4C,D demonstrates that AP’s frequency increases alongside the stimulation amplitude. Nevertheless, the amplitude of each next AP gradually decreases and APs are not observed during high-amplitude stimulation. It is noteworthy that stimulation amplitude insignificantly affects the amplitude of the first AP (Figure 4E). We have also found one interesting correlation: neurons that demonstrate HCN-currents (currents mediated by hyperpolarization-activated cyclic nucleotide-gated channels (HCN)) upon negative stimulation belong to the second group. These neurons are also distinguished by the high frequency of spike generation. Thus, the response of a particular neuron to paroxysmal depolarization depends on its electrophysiological properties and the amplitude of the input stimulus. Different stimulation protocols can confirm this correlation.

Figure 5A shows the responses of a representative neuron to three different stimulation protocols (rectangle, cosine, and triangle). It can be concluded that stimulation patterns and amplitude affect the response of neurons. The response to triangle stimulation is most similar to the reaction of this neuron to paroxysmal depolarization (Figure 5B). In contrast to cosine stimulation (smooth rise and decay), triangle stimulation demonstrates a steeper slope, likely more similar to paroxysmal depolarization. The experiments showing the effect of depolarization on PDS structure (Figure 2A, arrows 2 and 3) confirm this assumption since PDSs occurring at holding potential 0 mV are triangle-like. It may be suggested that the response to triangle 240 pA stimulation is more similar to the reaction to a paroxysmal shift because the depolarization amplitude corresponds to PMP_max_ (−30 mV) in this case (Figure 5(A′,B′)). Thus, using the triangle stimulation protocol, we can simulate paroxysmal shifts and predict the PDS structure for a particular neuron. Moreover, PDS clusters can be simulated with the modified triangle protocol consisting of consecutive stimuli. As shown in Figure 5C, 300 pA current injection results in a depolarization corresponding PMP_max_ value (−30 mV) and induces the generation of a PDS cluster consisting of four individual PDSs. The adaptation occurring after the first PDS and manifesting as a decrease in the number of APs is also observed in the case of triangle stimulation.

Thus, triangle stimulation seems to be a reliable and reproducible method of individual PDSs or PDS cluster induction. This protocol can be considered a convenient tool for studying paroxysmal activity at the cellular level since this approach allows the prediction of the response pattern to paroxysmal depolarization.

### 2.3. The Contribution of iGluRs in PDS Structure

As stated above, PDSs in neurons are generated in response to an external stimulus. We have investigated the contribution of each iGluR type in PDS structure since glutamate is the main excitatory neurotransmitter in hippocampal neuronal networks. Figure 6A, (Aʹ) demonstrate that the AMPAR antagonist (NBQX, 2 μM) completely suppresses bicuculline-induced [Ca^2+^]_i_ oscillations in all neurons. This effect is manifested at the level of individual neurons as suppression of PDS generation. The antagonist concentration was chosen based on the results of the previous experiments that demonstrated the blockade of AMPARs but not KARs by 2 μM of NBQX [13]. Thus, AMPARs play a critical role in paroxysmal activity, indicating that PDS generation is a cooperative network event. In turn, the NMDAR antagonist (D-AP5, 10 μM) decreased the amplitude and half-width of [Ca^2+^]_i_ oscillations (Figure 6B,(Bʹ)) but did not completely suppress them. The blockade of NMDARs did not affect resting potential, frequency of PDS cluster generation, or the number and frequency of PDSs in the cluster (Figure 6D). At the same time, D-AP5 reduced the amplitude of the first AP and PMP_max_/PMP_min_ values (Figure 6D). A decrease in the PMP_max_ value indicates that the amplitude of the input depolarizing stimulus is lower in this case. Therefore, membrane potential decays to more negative values before generating the next PDSs. As a result, the difference between PMP_min_ values for the first and penultimate PDSs in the cluster also decreases, and the patterns of individual PDSs become similar.

In the case of KARs, we used an antagonist of GluK1- and GluK3-containing receptors (primarily GluK1), UBP310 (Figure 7). As with the NMDAR antagonist, we found that the blockade of KARs decreased the amplitude and half-width of the [Ca^2+^]_i_ oscillations (Figure 7(A′)). However, UBP 310 reduced 1st AP amplitude and the number of PDSs in a cluster (Figure 7C) but did not affect the PDS structure (Figure 7(B′)). Thus, we have found that AMPARs are crucial in generating and maintaining paroxysmal activity. NMDARs enhance depolarizing stimulus, including due to Ca^2+^ influx, whereas KARs are likely implicated in the paroxysmal activity modulation, firstly affecting the number of PDSs in the cluster. It is noteworthy that although the blockade of NMDARs or KARs decreases the amplitude and half-width of calcium oscillations, the antagonists of these receptors insignificantly impact the oscillation frequency.

### 2.4. Agonists of Gi-Coupled Receptors Suppress Paroxysmal Activity

The occurrence of paroxysmal activity is considered a stage of epileptogenesis. Hence, searching for new targets for PDS suppression or its control is an actual task. Considering literature data and results of our previous studies, we have supposed that activation of G_i_-coupled receptors may suppress paroxysmal activity since these receptors modulate neurotransmission and intrinsic activity of neurons. As shown in Figure 8A–D, agonists of serotoninergic 5-HT_1_ (serotonin (5-hydroxytryptamine) 1A receptor) and A_3_ adenosine receptors did not suppress bicuculline-induced paroxysmal activity. In turn, WIN 55,212-2, an agonist of cannabinoid receptors, significantly decreased the [Ca^2+^]_i_ oscillation frequency at a concentration of 50 nM. In most experiments, the agonist of A_1_ adenosine receptors N_6_-cyclohexyladenosine (N_6_-CHA) suppressed paroxysmal activity at a concentration of 10 nM. Moreover, N_6_-CHA prevented the generation of the bicuculline-induced [Ca^2+^]_i_ oscillations at this concentration. Although preincubation of the cell cultures with WIN 55,212-2 reduced the frequency of [Ca^2+^]_i_ oscillations, we did not observe complete suppression in the case of this agonist. Thus, the agonist of A_1_ adenosine receptors was the most effective against bicuculline-induced paroxysmal activity. We found that N_6_-CHA (10 nM) completely suppressed and prevented PDSs and [Ca^2+^]_i_ oscillations in neurons.

## 3. Discussion

The paroxysmal activity of neurons can be induced in different ways, including the application of GABA(A)Rs antagonists, the elevation of extracellular K^+^ concentration, or a decrease in extracellular Mg^2+^ or Ca^2+^ concentration [14]. The shift of E/I balance towards excitation and further hypersynchronization of neuronal activity occurs in all these models. According to computer models, activation of several glutamatergic neurons is sufficient to induce paroxysmal activity in the whole network [15]. Interestingly, astrocytes are also involved in paroxysmal activity induction. It has been shown that PDSs in neurons are followed by depolarization and [Ca^2+^]_i_ elevations in astrocytes. Photolytic Ca^2+^ uncaging or IP_3_ infusion resulted in glutamate release from astrocytes and generation of PDSs in neighboring neurons [14,16]. Mathematical models confirm that astrocytes can induce or enhance paroxysmal activity [17]. Additionally, glutamate released by astrocytes is known to synchronize PDSs in neurons [16]. However, as we have previously shown, tonic glutamate release by astrocytes does not occur in rat neuron-glial hippocampal cultures grown in Neurobasal-A+B27 medium, even in the presence of bicuculline [18]. 

Hypersynchronization of neuronal networks is likely mediated by the synaptic and non-synaptic mechanisms, such as ephaptic interactions [19]. Glutamate secretion and further activation of postsynaptic NMDARs, AMPARs, and KARs can be considered an example of synaptic mechanisms. According to our experiments, AMPARs fulfill an essential role in maintaining paroxysmal activity since the blockade of these receptors results in complete suppression of PDSs in all neurons. Activation of NMDARs following activation of AMPARs provides additional ion influx, thus enhancing input neuronal stimulation and increasing the maximal amplitude of the depolarizing shift. We have first demonstrated the contribution of GluK1-containing KARs to paroxysmal activity. The blockade of KARs reduced the number of PDSs in the cluster insignificantly, affecting 1st AP amplitude and the number of PDSs in a cluster. This effect seems unexpected at first glance since expression of GluK1-containing KARs has been reported for GABAergic neurons [13,20]. Therefore, activation of these receptors should have induced GABA secretion and suppressed network activity, whereas the blockade, on the contrary, should have promoted excitation of the network. However, we observed the opposite effect. The peculiarities of the used model of paroxysmal activity may explain the described differences. As mentioned above, we used the competitive antagonist of GABA(A)Rs, bicuculline, to induce paroxysmal activity. The effects of GABA are mediated only by GABA(B)Rs in this case because bicuculline suppresses the activity of GABA(A)Rs. Although GluK1-containing receptors are predominantly expressed by GABAergic neurons, GluK1 homomers and heteromers also localize at pre- and postsynaptic terminals of glutamatergic neurons [20,21]. Thus, the KAR antagonist most likely suppresses glutamate release, reducing the number of PDSs in the cluster this way. 

In the close (self-contained) neuronal networks, extracellular stimulus inducing PDS generation has most likely a triangular shape. This conclusion has been drawn from the experiments shown in Figure 5. Most of the voltage-gated sodium and calcium channels are inactivated, whereas the conductivity of potassium channels is minimal at membrane potential close to 0 mV. The pattern of the depolarizing shift is primarily determined by iGluRs under these conditions [22,23]. The depolarizing signal is characterized by a fast rise followed by uniform, almost linear decay. This pattern reflects the changes in extracellular glutamate concentration. Barnes and coauthors have demonstrated that the dynamics of extracellular glutamate concentration upon powerful stimulation also has triangular-shape kinetics. Its duration corresponds to the duration of a single PDS [24]. Our experiments with triangle-shaped stimulation mimicking paroxysmal depolarization confirm this assumption. Based on this consideration, we can conclude that although the external depolarizing stimulus has a triangular shape and is mediated by the changes in glutamate concentration, the PDS pattern that we have observed in experiments is determined by the intrinsic properties of a neuron. For instance, voltage-gated sodium and high-threshold calcium channels are known to enhance initial iGluRs-mediated depolarization [25]. Additionally, Ca^2+^-activated potassium channels and voltage-gated L-type calcium channels modulate the amplitude and duration of a PDS [9,25]. In turn, the potassium channels shape the hyperpolarizing component of PDS [25,26]. Intrinsic electrophysiological properties of neurons, particularly their ability to generate high-frequency spikes, determine the number of APs occurring during PDS. The number of these APs depends on the amplitude of the depolarizing stimulus because the number of spikes that a particular neuron can generate is determined by membrane potential. Each PDS is terminated due to the activity of the potassium channels and the depletion of extracellular glutamate. The glutamate concentration in a synaptic cleft depends on the intensity of its secretion from a presynaptic terminal and the rate of uptake by neuronal and especially astrocytic glutamate transporters. Moreover, the GABA transporters, which compensate for the excitatory action of glutamate by governing GABA concentration, are also crucial in this case. Although the volume of ‘extracellular space’ in 2D cell cultures significantly differs from the native brain (large volume of the extracellular medium above the monolayer), the fundamental mechanisms underlying the maintenance of synaptic glutamate/GABA concentration are similar in both models. The similarity of the kinetics of glutamate concentration in the brain under pathological conditions and the pattern of paroxysmal depolarization (triangle shape) in our model may indirectly confirm this hypothesis. Each new PDS during the cluster occurs when membrane potential in the group of initiative neurons is sufficient to overcome the AP threshold, and the generated spikes trigger simultaneous synchronous glutamate release. It has been shown that membrane hyperpolarization followed by slow repolarization occurs in the initiative neurons after PDS [10,15]. The depolarization rate obviously determines the interval between PDSs. It may be possible that this rate depends on the intrinsic properties of neurons and external factors, such as the rate of restoration of extracellular ion balance, particularly K^+^ concentration, which is regulated by the activity of astrocytes [14,15]. 

The mechanisms underlying the integration of several PDSs into clusters remain unknown. Our experiments show that duration and PMP_max_ values are similar for individual PDSs in the cluster. However, the intervals between consecutive PDSs increase during the cluster. We did not find reliable markers in electrophysiological recordings demonstrating whether a particular PDS is the last or whether others can follow it. Considering that the external input stimulus causes each PDS in the cluster, it may be suggested that all PDSs following the first occur in response to glutamate release. Thus, an unrevealed factor promotes several powerful glutamate effluxes during the short interval. High-frequency recordings (Figure 1) demonstrate that PDSs are followed by the quasi-synchronous [Ca^2+^]_i_ elevations consisting of some peaks. The number of peaks corresponds to the number of PDSs in the cluster; therefore, this number is similar for all neurons, at least in a viewfield. It should be noted that the [Ca^2+^]_i_ level remains high in the periods between the peaks. In this regard, a significant increase in [Ca^2+^]_i_ during the first PDS might be considered as the factor promoting repetitive glutamate release and consecutive generation of PDSs. This hypothesis is supported by the experiments where the blockade of L-type voltage-gated calcium channels profoundly reduces the number of PDSs in the cluster, insignificantly affecting the structure of individual PDSs [10]. Similarly, the blockade of GluK1-containing KARs that may be Ca^2+^-permeable [27] also reduces the amplitude of [Ca^2+^]_i_ oscillations and the number of PDSs in the cluster. However, the NMDAR blockade, followed by a decrease in the amplitude of the oscillations, did not reduce the number of PDSs in the cluster. Therefore, the pathway of Ca^2+^ inflow plays a critical role in this case. NMDAR-mediated Ca^2+^ entry first affects the amplitude of the depolarizing stimulus and insignificantly impacts the generation of the subsequent PDSs. In turn, Ca^2+^ entry through the other pathways, such as L-type voltage-gated calcium channels, promotes glutamate secretion or increases neuronal excitability, thus decreasing the activation threshold. Nevertheless, this mechanism needs to be better recognized, and further studies are required.

### G_i_-Coupled Receptors 

Here we show that the activation of some subtypes of G_i_-coupled receptors suppresses bicuculline-induced paroxysmal activity. In particular, the agonists of CBRs and A_1_Rs significantly reduced the frequency of [Ca^2+^]_i_ oscillations or even completely suppressed paroxysmal activity. Moreover, preincubation with the A_1_R agonist abolished the induction of the bicuculline-induced oscillations. We did not observe a similar effect in the case of the CBR agonist. This fact may indicate that the mechanisms underlying the inhibitory action of the A_1_R and CBR agonists are different. 

Different signaling cascades can mediate the effects of A_1_Rs and CBRs in our experiments [28,29]. These receptors have been found at pre- and postsynaptic terminals and even in astrocytes [30,31,32]. Stimulation of A_1_Rs and CBRs results in hyperpolarization of neurons mediated by different potassium channels, including GIRK, K_ATP_, and SK channels [15,16,28]. Furthermore, a decrease in the cAMP level caused by adenylyl cyclase inhibition attenuates neurotransmission and reduces NMDAR and AMPAR conductance [28,32,33]. It is logical to assume that stimulation of both receptors suppresses neuronal activity in the same way; however, the activation of CBRs in contrast to A_1_Rs did not prevent the induction of paroxysmal activity. Hence, it can be proposed that the effects mediated by these receptors vary and depend on other factors, such as expression profile or localization. 

The ability of A_1_R agonists to prevent the bicuculline-induced paroxysmal activity even at nanomolar doses indicates the potential perspective of using these drugs for preventive therapy in patients with epilepsy. As known, PDSs occur in the periods between the seizures or precede them [26]. Paroxysmal activity registered with EEG can be considered prologue and a diagnostic marker for epilepsy. The epidemiological studies performed after the war in Vietnam and Croatia demonstrated that epilepsy developed 10–15 years after the penetrating brain wound [34,35]. According to other studies, paroxysmal activity was observed in 80% of people 24 h after the penetrating brain wound [36]. Based on these reports, we suppose that prevention or suppression of paroxysmal activity may abolish future epilepsy development. 

## 4. Materials and Methods

### 4.1. Preparation of Hippocampal Cell Culture

Neuron-glial cell cultures were prepared in the same manner as described previously [10,13]. Wistar pups (P0-2) were euthanized with deep-inhaled anesthesia and decapitated. Extracted brains were transferred into a plastic Petri dish (d = 60 mm) filled with cold Versene solution. The separated hippocampus was carefully minced with scissors and treated with 1% trypsin solution for 10 min at 37 °C and with constant shaking. The minced tissue was washed twice with cold Neurobasal-A medium to inactivate trypsin. Then, tissue fragments were triturated by slowly passing through a 1 mL pipette tip. The debris was carefully removed with a 200 µL pipette tip, and the obtained single-cell suspension was centrifuged for 3 min at 2500 rpm. After that, the supernatant was removed, and the cell pellet was resuspended in Neurobasal-A medium supplemented with B27 (2%), 500 µM glutamine, and penicillin-streptomycin (1:100). The cells were seeded on polyethyleneimine-coated round (d = 25 mm) coverglasses placed in cell culture Petri dishes (d = 35 mm). The cultures were grown at 37 °C in a humidified atmosphere (humidity ≥ 90%) containing 5% CO_2_ and were used in experiments at 13–14 DIV (days in vitro).

### 4.2. [Ca^2+^]_i_ Imaging

In most experiments, the fluorescent Ca^2+^-sensitive probe Fura-2 AM was used to evaluate the changes in intracellular Ca^2+^ concentration ([Ca^2+^]_i_). We used Fluo-4 AM instead of Fura-2 in particular experiments with high-speed acquisition (Figure 1D). All the fluorescent measurements were performed in Hank’s balanced salt solution (HBSS) consisting of (in mM): 136 NaCl, 3 KCl, 0.8 MgSO_4_, 1.25 KH_2_PO_4_, 0.35 Na_2_HPO_4_, 1.4 CaCl_2_, 10 glucose, and 10 HEPES; pH 7.35. Fura-2 stock solution was dissolved in HBSS to a final concentration of 3 µM. The cells were incubated with the probe for 40 min at 28 °C and washed thrice. Two-channel series of images were recorded using an inverted motorized fluorescent microscope Leica DMI 6000B (Leica Microsystems, Wetzlar, Germany) with a CCD camera Hamamatsu C9100 (Hamamatsu Photonics K.K., Hamamatsu City, Japan). External filter wheel with excitation filters BP340/30 and BP387/15, and internal FU-2 filter cube (dichroic mirror 72100bs, emission filter HQ 540/50 m) (Leica Microsystems, Wetzlar, Germany) were used for ratiometric fluorescent measurements. In the case of Fluo-4, the incubation protocol was similar to that of Fura-2. The final concentration of the probe in the working solution was 5 µM. Fluo-4 fluorescence was excited and registered using L5 filter cube consisting of excitation filter BP480/40, dichroic mirror 505 nm, and emission filter BP 527/30.

The images were analyzed with ImageJ (NIH, Bethesda, MD, USA) software, following the previously reported protocol [25]. Changes in [Ca^2+^]_i_ are presented as 340/387 ratio for Fura-2 and as ΔF/F0 for Fluo-4. To identify neurons, short-term KCl applications were made before or after the experiments (not shown in panels) [18]

### 4.3. Electrophysiological Measurements

#### 4.3.1. Equipment and Solutions

All whole-cell recordings of membrane voltage were performed at 28 °C in HBSS solution; the composition was described in Section 2.2. We used borosilicate glass capillaries with filament (Sutter Instrument, Navatto, CA, USA) for micropipette pulling. Micropipettes were pulled with a vertical micropipette puller Narishige PC-100. The next intrapipette solution was used in all experiments (in mM): 10 KCl, 125 K-gluconate, 1 MgCl_2_ × 6H_2_O, 0.25 EGTA, 10 HEPES, 2 Na_2_-ATP, 0.3 Mg-ATP, 0.3 Na-GTP, 10 Na_2_-phosphocreatine (pH 7.2; adjusted with 1 M KOH). Data were recorded with Axopatch 200B amplifier, filtered, and digitized with a low-noise data acquisition system Axon DigiData 1440A digitizer (Molecular Devices, San Jose, CA, USA) at a sampling rate of 10 kHz. 

#### 4.3.2. Stimulation Protocols 

To induce a single PDS, we used the following stimulation protocol (Figure 5A): 100 ms resting membrane potential recording → step current injections with delta level 30 pA (15 steps) and 300 ms duration → 2 s resting membrane potential recording → cosinusoidal stimulation with 250 ms duration and 4 Hz train rate → 2 s resting membrane potential recording → triangle stimulation with 250 ms duration, train rate 4 Hz and pulse width 10 ms. In the case of the PDS cluster induction (Figure 5C), we used triangle stimulation (10 steps with 50 pA step size): 2 s resting membrane potential recording → triangle stimulation with 1 s duration, train rate 4 Hz, and pulse width 10 ms → 2 s resting membrane potential recording. All stimulations and recordings were performed using pCLAMP 10 software (Molecular Devices, San Jose, CA, USA).

### 4.4. Statistical and Data Analysis

Origin Pro 2021 version 9.8.0.200 was used for graph creation and analysis (OriginLab, Northampton, MA, USA). Electrophysiological data were analyzed using ClampFit 10 software (Molecular Devices, San Jose, CA, USA). The Shapiro–Wilk test (*p* > 0.05) was used to evaluate the normality of data distribution since the sample size was *n* < 15. Normality tests performed with Origin Pro showed that all datasets were normally distributed. Therefore, we used parametric tests. The differences were analyzed with (GraphPad Software, San Diego, CA, USA) paired or unpaired t-tests (two-tailed) and one-way ANOVA (one-tailed) followed by Dunnett’s multiple comparisons tests using GraphPad Prism 8. Significance levels are defined with a *p* value less than 0.05. The number of independent cell culture preparations used in the experiments is marked as “N,” whereas the total number of analyzed cells (in all used preparations) is marked as “*n*.” As a rule, cell culture preparations from 2 to 3 different animals were used in experiments. 

The following parameters for box plots in the figures were defined: dimensions—75th (top) and 25th (bottom) percentiles; line—median; whiskers—minimal and maximal values. Values are presented as mean ± SD. 

### 4.5. Reagents

The reagents that were used in the experiments are listed below. (1) *Sigma-Aldrich, Saint Louis, MO, USA*: Poly(ethyleneimine) solution (Cat. no. P3143), penicillin–streptomycin (Cat. no. P4333), L-Glutamine (Cat. No G85402). (2) *Life Technologies, Grand Island, NY, USA*: B-27 supplement (Cat. no. 17504044), Trypsin 2.5% (Cat. no. 15090046). (3) Molecular Probes, Eugene, OR, USA: Fura-2 AM (Cat. no. F1221). (4) *Tocris Bioscience, Bristol, UK*: UBP 310 (Cat. no. 3621), 5-Nonyloxytryptamine oxalate (Cat. No. 0901), WIN 55,212-2 mesylate (Cat. no. 1038) (5) *Alomone Labs, Jerusalem, Israel*: D-AP5 (Cat. no. D-145), NBQX (Cat. no. N-185). (6) *Cayman Chemical, Ann Arbor, MI, USA:* Bicuculline (Cat. no. 11727), HEMADO (Cat. No. 21015); (7) *AppliChem, Darmstadt, Germany:* EDTA (Cat. no. A5097), EGTA (Cat. no. A-0878). (8) *Dia-M, Moscow, Russian Federation:* HEPES (Cat. no. 3350). (9) *Abcam, Cambridge, UK:* N6-Cyclohexyladenosine (Cat. no. ab120472). (10) *Paneco, Moscow, Russian Federation:* Neurobasal-A medium.

## 5. Conclusions

Our experiments demonstrate that PDSs, which considered the cellular correlate of interictal spikes, are caused by network activity. We have shown that PDS generation is associated with the activity of iGluRs (primarily AMPARs), whereas the mechanism underlying the integration of single PDSs into the cluster remains unknown and further studies are needed. The obtained data open avenues to further understanding epileptogenesis mechanisms and the development of new therapeutic approaches. Of course, the used 2D neuron-glial cell cultures significantly differ from the native brain tissue, and the obtained results have to be carefully extrapolated to in vivo studies. However, the similarity in the paroxysmal activity patterns registered in in vitro and in vivo studies gives us hope that the fundamental mechanisms underlying PDS generation and maintenance are similar, and our findings may be valid for the whole brain. Regarding epilepsy treatment, we have shown that A_1_ adenosine receptor agonists seem promising as antiepileptic drugs since their application not only suppresses but also prevents paroxysmal activity. In turn, the exact downstream signaling mechanism mediating the ‘antiepileptic’ action of A_1_ adenosine receptor agonists is unknown and requires further detailed investigation.

## Figures and Tables

**Figure 1 ijms-24-10991-f001:**
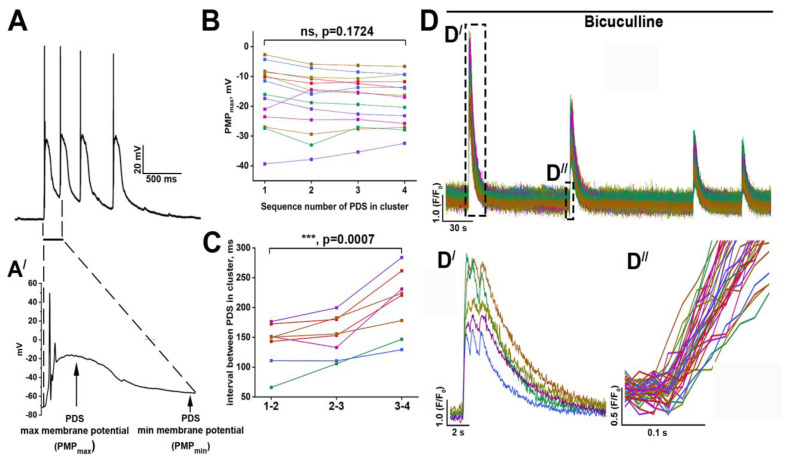
PDS cluster and synchronous [Ca^2+^]_i_ oscillations in neurons. (**A**,**A′**) Typical PDS cluster (**A**) and enlarged the first PDS (**A′**); (**B**) Diagram showing the PMP_max_ values for consecutive PDSs during the cluster; (**C**) Changes in intervals between individual PDSs in the cluster. In the case of figures B and C, the differences were analyzed with one-way ANOVA (analysis of variance) followed by Dunnett’s multiple comparison test. *p* < 0.001 (***). (**D**,**D′**,**D″**)High-frequency registration of [Ca^2+^]_i_ transients were observed during bicuculline-induced paroxysmal activity (10 μM). *n* = 300, *n* = 3. Different colors of the curves in all figures demonstrate different representative neurons analyzed in the experiments.

**Figure 2 ijms-24-10991-f002:**
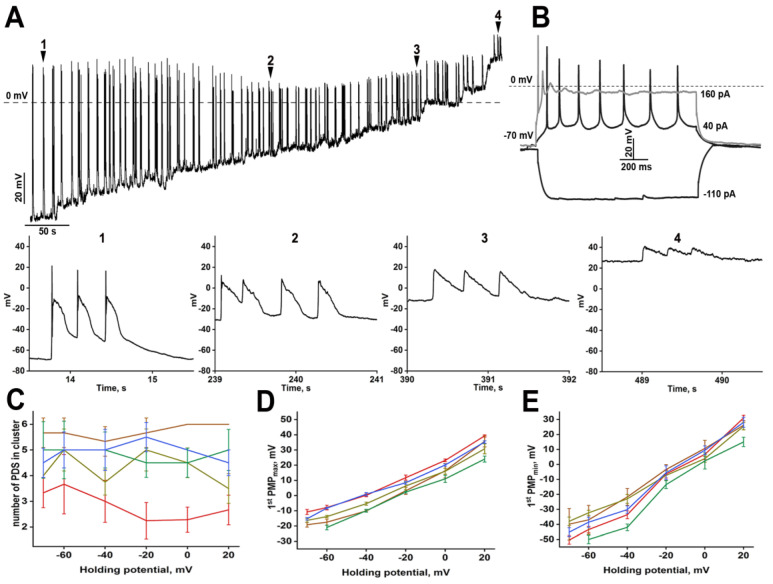
(**A**) The effect of membrane potential depolarization on PDS structure. Enlarged individual PDSs are shown below panels (**A**) and (**B**). The numbers in panels with enlarged PDSs correspond to the numbers in panel (**A**); (**A**) The response of the neuron from panel (**A**) to gradual hyperpolarizing and depolarizing stimuli; (**C**) The dependence of the number of PDSs in the cluster on the holding membrane potential. Each curve corresponds to the number of PDSs in a cluster averaged by one representative neuron; (**D**,**E**) The dependence of PMP_max_ (**D**) and PMP_min_ (**E**) on the holding membrane potential. *n* = 5; *n* = 5 for panels (**D**,**E**). Each trace in panels (**C**–**E**) corresponds to one representative neuron. Different colors of the curves in figures (**D**,**E**) demonstrate different representative neurons analyzed in the experiments.

**Figure 3 ijms-24-10991-f003:**
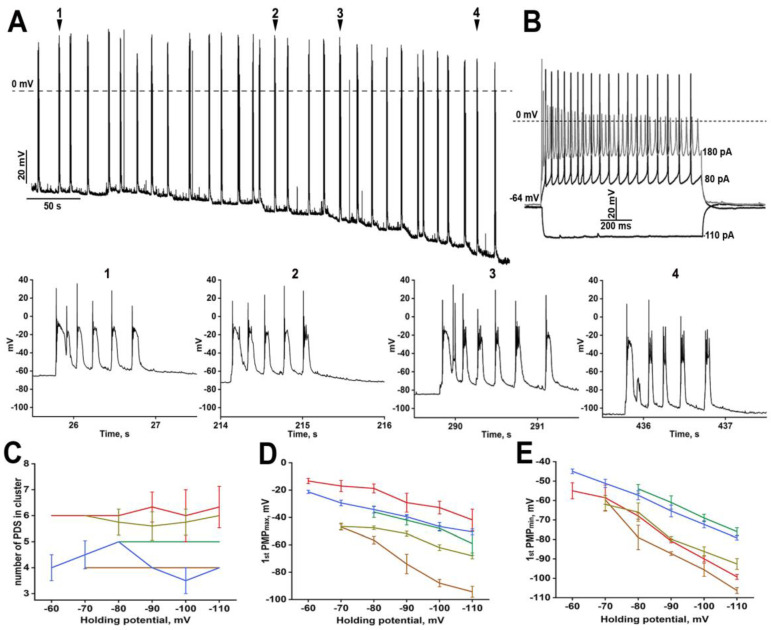
(**A**) The effect of membrane potential hyperpolarization on PDS structure. Enlarged individual PDSs are shown below panels (**A**) and (**B**). The numbers in panels with enlarged PDSs correspond to the numbers in panel (**A**); (**B**) The response of the neuron from panel (**A**) to gradual hyperpolarizing and depolarizing stimuli; (**C**) The dependence of the number of PDSs in the cluster on the holding membrane potential. Each curve corresponds to the number of PDSs in a cluster averaged by one representative neuron; (**D**,**E**) The dependence of PMP_max_ (**D**) and PMP_min_ (**E**) on the holding membrane potential. *n* = 5, *n* = 5 for panels (**C**–**E**). Each trace in panels (**C**–**E**) corresponds to one representative neuron. Different colors of the curves in figures (**C**–**E**) demonstrate different representative neurons analyzed in the experiments.

**Figure 4 ijms-24-10991-f004:**
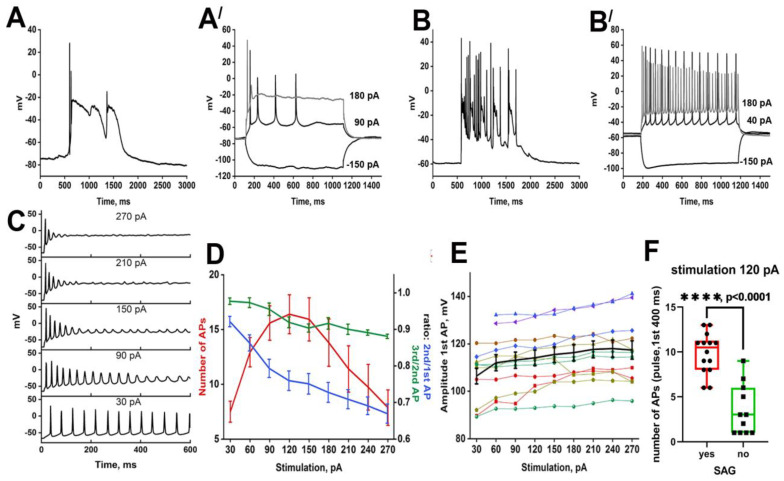
PDS structure and responses of two representative neurons (**A**,**Aʹ** and **B**,**Bʹ**) to different stimulations; (**C**) The response of a representative neuron to gradual depolarizing current injections; (**D**) Dependence of the number of APs, 1st AP amplitude/2nd AP amplitude, and 1st AP amplitude/3rd AP amplitude ratios on injection current amplitude. The mean values are shown in graphs; (**E**) The dependence of the first AP amplitude on stimulation amplitude; (**F**) The comparison of the number of APs during the first 400 ms of 120 pA current injection in neurons with and without HCN-currents. Student *t*-test. *p* < 0.0001 (****). Different colors of the curves in figures (**D**,**E**) demonstrate different representative neurons analyzed in the experiments.

**Figure 5 ijms-24-10991-f005:**
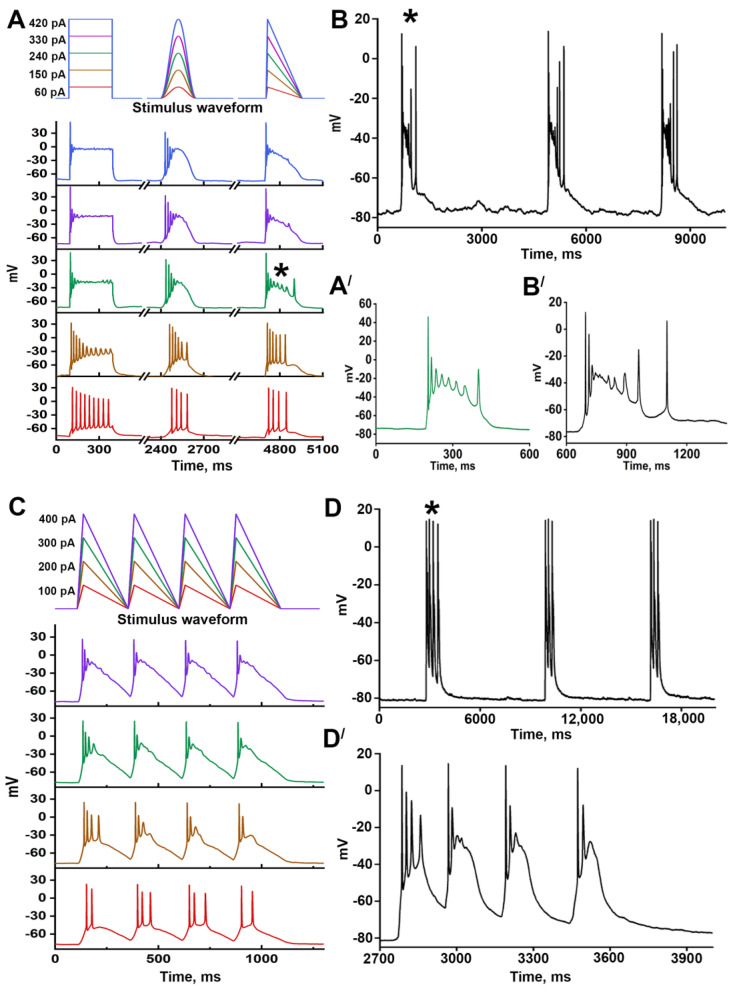
The response of neurons to different stimulation protocols: (**A**,**C**) Stimulation protocols (**top** panels) and corresponding responses of neurons (**bottom** panels); different colors of the curves correspond to different current injections. (**B**,**D**) Recordings of paroxysmal activity of representative neurons, where responses are shown in panels (**A**,**C**), respectively; (**A′**,**B′**) Enlarged images of individual PDSs marked with asterisks (*) in panels (**A**,**B**), respectively; (**D′**) Enlarged image of the PDS marked with the asterisk in panel (**D**).

**Figure 6 ijms-24-10991-f006:**
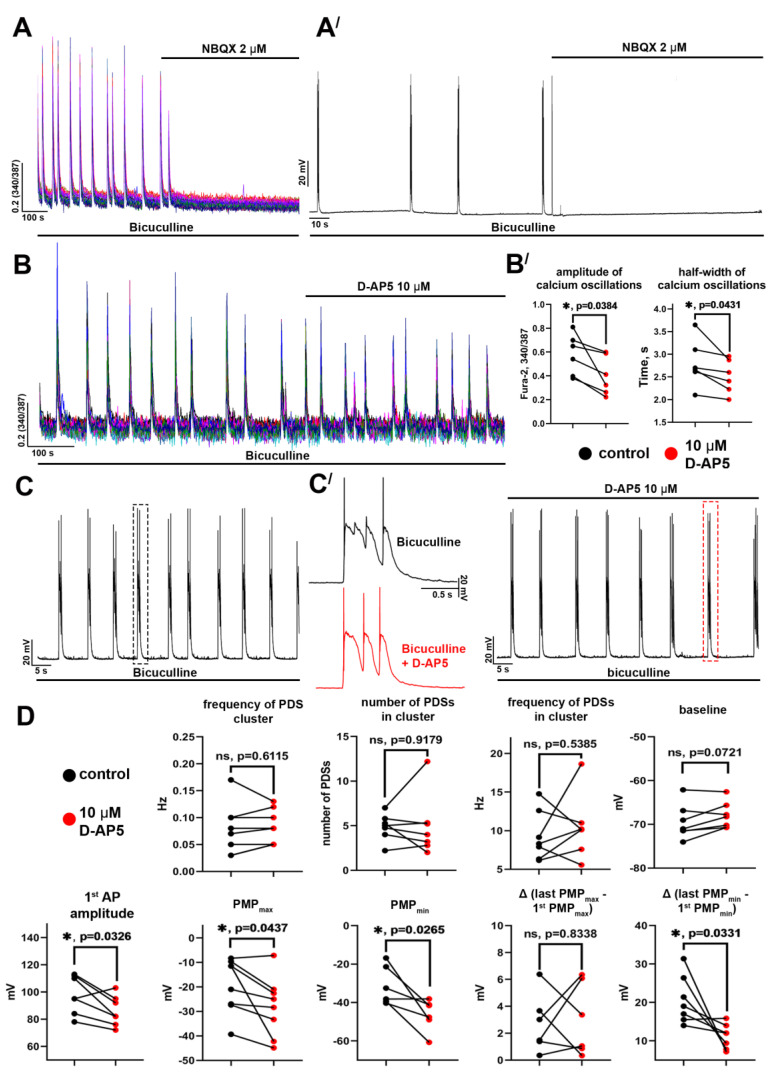
The role of AMPA and NMDA receptors in paroxysmal activity: (**A**,**A′**) The effect of AMPAR antagonist NBQX on bicuculline-induced synchronous [Ca^2+^]_i_ oscillations (**A**) and PDSs (**A′**); (**B**) The impact of NMDAR antagonist D-AP5 on bicuculline-induced synchronous [Ca^2+^]_i_ oscillations; (**B′**) The diagrams show the amplitude and frequency of [Ca^2+^]_i_ oscillations before (black dots) and after (red dots) D-AP5 application; paired *t*-test, *p* < 0.05 (*); (**C**) The effect of D-AP5 on the bicuculline-induced PDSs; (**C′**) The enlarged PDSs marked with a dashed line in panel (**C**); (**D**) The diagrams showing the values of different parameters of single PDSs and PDS clusters before (black dots) and after (red dots) D-AP5 application; paired *t*-test, *p* < 0.05 (*).

**Figure 7 ijms-24-10991-f007:**
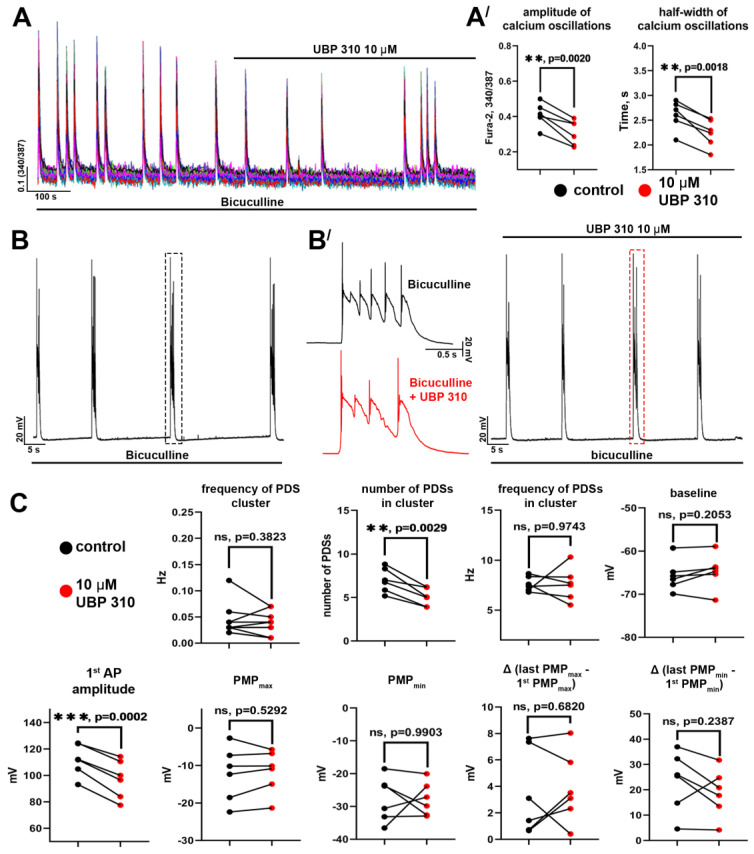
The role of GluK1-containing KARs in paroxysmal activity: (**A**) UBP310, an antagonist of GluK1/GluK3-containing KARs, affects bicuculline-induced synchronous [Ca^2+^]_i_ oscillations; (**A′**) The diagrams showing the values of the [Ca^2+^]_i_ oscillation amplitude and frequency before (black dots) and after (red dots) UBP310 application; paired *t*-test, *p* < 0.01 (**); (**B**) The effect of UBP 310 on the bicuculline-induced PDSs; (**B′**) The enlarged PDSs marked with a dashed line in panel (**B**); (**C**) The diagrams showing the values of different parameters of single PDSs and PDS clusters before (black dots) and after (red dots) UBP310 application; paired *t*-test, *p* < 0.01 (**), *p* < 0.001 (***).

**Figure 8 ijms-24-10991-f008:**
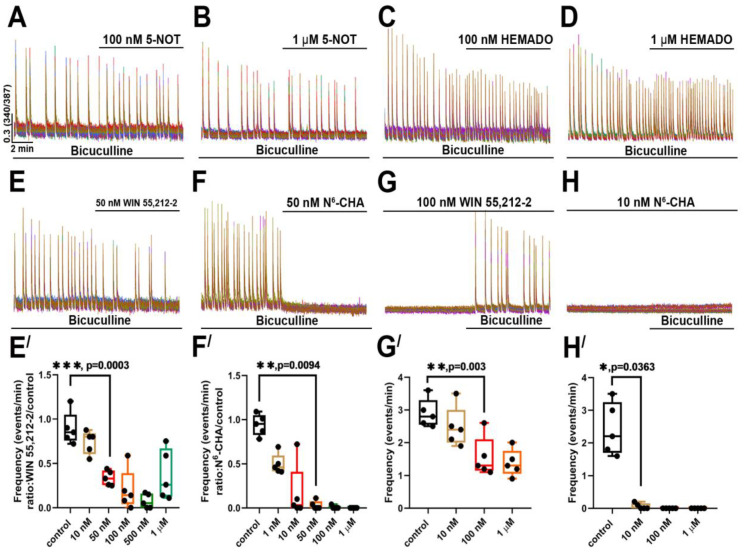
The effect of agonists of G_i_-coupled receptors on bicuculline-induced [Ca^2+^]_i_ oscillations: (**A**–**F**) Application of agonists of (**A**,**B**) 5-HT_1_ (5-NOT, 100 nM and 1 μM), (**C**,**D**) adenosine A_3_ (HEMADO 100 nM and 1 μM), (**E**) cannabinoid (WIN 55,212-2, 50 nM), (**F**) adenosine A_1_ (N_6_-CHA, 50 nM) receptors during bicuculline-induced paroxysmal activity; (**G**,**H**) Induction of paroxysmal activity in the presence of (**G**) cannabinoid (WIN 55,212-2, 100 nM) and (**H**) adenosine A_1_ receptor agonists (N_6_-CHA, 10 nM); (**E′**,**F′**) Diagrams showing the dependence of the frequency of calcium oscillations on (**E′**) WIN 55,212-2 and (**F′**) N_6_-CHA concentration; one-way ANOVA, *p* < 0.01 (**), *p* < 0.001 (***); (**G′**,**H′**) Dependence of frequency of calcium oscillations induced in the presence of different (**G′**) WIN 55,212-2 and (**H′**) N_6_-CHA concentrations; one-way ANOVA, *p* < 0.05 (*), *p* < 0.01 (**).

## Data Availability

The data supporting this study’s findings are available from the corresponding author upon reasonable request.

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
