# Peer review of "Of the Mechanisms of Paroxysmal Depolarization Shifts: Generation and Maintenance of Bicuculline-Induced Paroxysmal Activity in Rat Hippocampal Cell Cultures"

_ijms, 2023, doi:10.3390/ijms241310991_

Round 1

Reviewer 1 Report

 The manuscript by Laryushkin,D P  et. al., titled “Of the mechanisms of paroxysmal depolarization shifts: generation, maintenance, suppression” is an interesting study that can be considered for publication with minor edits.

 Minor:

 1.     Figure 1, 2 and 3 please increase the font size of X and Y axis or increase the size of the whole figure

2.     Figure 1. A   EP data show Bicuculline treatment similar Fig1.D Ca2+ imaging

3.     Figure 2 A&B and in Figure 3 A&B please insert electrophysiology protocol below the traces

4.     Figure 2 B and Figure 3B move the scale bar outside the traces

5.     Figure 2 C, D, E and Figure 3 C,D,E   please mention what the colored line represents in legends  

6.     Figure 2. (A) The effect of membrane potential depolarization on PDS structure. Enlarged individual 131 PDSs are shown below the panels A and B.

Author Response

The manuscript by Laryushkin,D P  et. al., titled “Of the mechanisms of paroxysmal depolarization shifts: generation, maintenance, suppression” is an interesting study that can be considered for publication with minor edits.

 Minor:

  1. Figure 1, 2 and 3 please increase the font size of X and Y axis or increase the size of the whole figure

The figure size corresponds to the MDPI standards.

  1. Figure 1. A   EP data show Bicuculline treatment similar Fig1.D Ca2+imaging

Unfortunately, we have not understood this comment. Could you clarify this point, please?

  1. Figure 2 A&B and in Figure 3 A&B please insert electrophysiology protocol below the traces.

We did not use any standard protocol. Instead, we gradually manually change the holding potential.

  1. Figure 2 B and Figure 3B move the scale bar outside the traces

The scale bars have been moved.

  1. Figure 2 C, D, E and Figure 3 C,D,E   please mention what the colored line represents in legends  

Each trace in panels C-E corresponds to one representative neuron. The necessary remarks have been added into the legends.

  1. Figure 2. (A) The effect of membrane potential depolarization on PDS structure. Enlarged individual 131 PDSs are shown below the panels A and B.

Unfortunately, we have not understood this comment. Could you clarify this point, please?

Reviewer 2 Report

If the title of the manuscript is taken at the face value, Laryushkin et al wish to communicate their findings on the generation, maintenance and possible control (suppression?) of a particular physiological phenomena with links to the mechanisms of epilepsies. Their efforts are commendable because, despite many advances in the field, etiology of epilepsy remains poorly understood and, in many patients, it remains an intractable and, essentially idiopathic, disease. I have several issues with the manuscript, especially how it is presented and how the data are evaluated and interpreted. These involve mainly the title, Introduction, Discussion but include also more fundamental problems such as whether the cultured neurons are a suitable model and how far can the conclusions be trusted (cf. ## 14 and 17)  Authors should carefully consider what is or what is not new (and credible) in their study and reformulate their discussion and conclusions accordingly. I am addressing the main points under the specific comments and queries below:

1.     At the very least, Title should indicate the experimental model (cultured hippocampal cells) and the experimental animal (rat).

2.     Abstract sounds too general, not sharp enough. I would start by specifying what precise “pivotal role” AMPA receptors are proposed to play and by stating which particular mGluR agonists/antagonists were used. This is mainly to make the Abstract more attractive to potential readers browsing the data bases and, hopefully, increasing the impact, if the paper is published.

3.     Introduction is too long, almost review-like and, at the same time, leaves some important information out. It should emphasize the points which prompted the study. Let’s not forget that Glutamate, GABA, AMPA, kainate receptors (and more) in the mechanisms of epilepsy would be an old hat to many readers who may be looking for a fresh approach and new information.

4.     Line 45: Yes, indeed, there is strong evidence that even GABA itself can act as a maturation factor. See eg. Wolff et al.  Adv Exp Med Biol 181 215-239, 1984 (doi: 10.1007/978-1-4684-4868-9_17.) and several more papers from the same laboratory in the 1980’s and 1990’s. One must wonder whether a GABA deficit or malfunction of GABA systems during development do not predispose to epilepsy. Maybe that the topic should be revisited.

5.     Line 54: “assumption is confirmed”, not the best form. Try – “this hypothesis has been tested” or “this proposal has been further buttressed” or simply “this is further supported by” see also L. 378

6.     Lines 61/62 ”myriads.. ..mechanisms,” This sounds very clumsy, hard to understand. Try “..in a large number of neurons firing synchronously and involves many distinct mechanisms such as communications..”

7.     Line 68: “predominantly localize”, not the happiest formulation. Kainate receptors, to my knowledge, occur both pre- and postsynaptically; see Falcon-Moya R et al: Front. Mol. Neurosci., 22 June 2018 Sec. Brain Disease Mechanisms Volume 11 – 2018 https://doi.org/10.3389/fnmol.2018.00217. The proposal that kainate receptors act on neurotransmitter release has been considered for a long time, too, but it is probably not their only role.

8.     Line 94: Introduction should summarize the state of the art in the field, formulate a hypothesis to be tested and state how it will be done. In other words, readers expect more of what “we intend to examine/test” rather than what “we have demonstrated”, the latter being better left for Conclusion. The findings belong to Results and their significance to Discussion. As it stands, we have covered nearly one hundred lines and, so far, we have no word on what exactly we want to do and what experimental model we shall be using, not to mention the experimental animal.

9.     Lines 95/96: Historically, bicuculline is certainly the most important GABA antagonist and it was instrumental in identifying GABA as the main inhibitory neurotransmitter (Neurochem Int 35 269-280, 1999 but it has problems with solubility and stability as well as specificity (e.g. Breuker E, Johnston GAR, J Neurochem 25 903-904, 1975). Gabazine (SR95531) seems to be a more specific GABA antagonist (there should not be many inhibitory gabazine-sensitive glycine receptors in the current experimental model) and is much easier to handle. Have you tried it?

10.   Line 128: “the next”, better “the following”

11.   Line 166: “in dependence”, better “depending on”

12.   Line 307: “non-synaptic mechanisms” Gap junctions form "electrical synapses" so the communication by Gap Junctions is, to be physiologically precise, not "non-synaptic" (Pereda AE, Nature Reviews Neuroscience  15, 250–263, 2014).

13.   Line 317: “..we have shown.. ..promising antiepileptic drugs, ..” It was actually shown years ago by others (reviewed, e.g., in Int Rev Neurobiol 119, 233-255, 2014, doi: 10.1016/B978-0-12-801022-8.00011-8). So, what is really new in your conclusion?

14.   Discussion: There is a potentially very important element missing in the evaluation of the data. Discussion aims mainly at receptors and completely disregards the presence of any neurotransmitter inactivation system. I mean the glutamate transporters such as GLT1 or GLAST, present in astrocytes (recent comprehensive reviews: Verchratsky & Niedergaard, Physiol Rev 98, 239-389, 2018; Andersen & Schousboe J Neurochem doi: 10.1111/jnc.15811, 2023). Yet astrocytes are obviously present in the cell culture and the neurotransmitter transport may play a very significant part in the formation and modification of the synaptic events (and in the mechanisms of epilepsies, Tanaka et al, Science 276 1699-1702, 1997). There is a more specific problem; in the cell culture, neurotransmitter may rapidly diffuse away from the site of action at the synapses which are exposed to a very large “extracellular space” i.e. the space above the monolayer culture filled with medium (thus creating a steep neurotransmitter concentration gradient above the “two-dimensional” cell culture). The role of transporters may appear, under such circumstances, limited; one could be tempted to posit (misleadingly) neurotransmitter release (at the expense of neurotransmitter re-uptake) as the most important determinant of the neurotransmitter concentrations at the synapse. In contrast, the extracellular space in vivo is much smaller, diffusion is hindered and not fast enough (Sykova & Nicholson Physiol Rev 88 1277-1340, 2008); the transport may then become much more important for shaping the synaptic activity.  This can be illustrated by somewhat conflicting conclusions drawn from two different types of experiments, both looking at the role of transporters, but using different models; Mennerick & Zorumski  J Neurosci 15 3178-3192, 1995, cell cultures) question the significance of transporters at the synapse while Diamond & Jahr J Neurophysiol 83 2835-2843, 2000, hippocampal slices preserving the three-dimensional structure of the tissue) ascribe the glutamate transporters a direct role in the synaptic function. Authors should, at the very least, remind the readers of a potential role of glutamate (and GABA) transporters in modifying the epilepsy-related mechanisms, at least in vivo or in alternative experimental models. This would not necessarily invalidate their present conclusions, just limit them to the current experimental model thus avoiding possible future misunderstandings and/or false claims.  

15.   One other comment related to the above point: Kainate also inhibits glutamate transport (Johnston et al, J Neurochem 32, 121-127, 1979) and authors may wish to acknowledge it.

16.   It is not entirely clear to me why the glutamate release from astrocytes has to be discussed in such a detail, particularly when it was not directly investigated. I accept that the photolytic liberation of Ca2+ from its caging may have been linked to the release of glutamate from astrocytes in other studies (ref. 27 and 29) but it has not been looked at in the present experiments. In any case, how would you tell the difference between neuron-originating and astrocyte-originating glutamate? Additionally, glutamate could be released from astrocytes in a non-vesicular manner by reversed transport process (Szatkowski et al Nature 348 443-446, 1990). This sometimes happens in mammalian glia, too, particularly under the conditions of ischemia, or, generally, when the cells are metabolically compromised (could it happen during prolonged epileptic discharge?).

17.   The finding that the cultured neurons display, individually and collectively, types of behaviour reminiscent of phenomena associated with epilepsies is, of course, encouraging. It would be good though, to mention (possibly in Conclusion or in a separate section on ”Limitations of the Study”) that cultures grown in Petri dish (and/or on coverslips) in the presence of a potential convulsant (penicillin) may not faithfully represent three-dimensional structure of brain (hippocampal) tissue (cf. # 14 above) and the present conclusion need to be further tested using additional experimental models. This might appear to devalue the outcome of the study somewhat, but it could pre-empt hostile criticism and make the final arguments (suitably modified) more credible.

18.   MS would benefit from a list of abbreviations. Supposing I want to know the meaning of “HCN” and “SAG” (Fig. 4 and lines 181 and 192)?

Only minor corrections required, as shown in the above comments. Also spell and grammar check of any future version.

Author Response

The authors sincerely thank the Reviewer for the detailed analysis and criticism of our study. We have divided all the listed comments into some sections. Our responses are highlighted with green color.

Title and abstract

  1. At the very least, Title should indicate the experimental model (cultured hippocampal cells) and the experimental animal (rat).

We have revised the title. The current version is: “Of the mechanisms of paroxysmal depolarization shifts: generation and maintenance of bicuculline-induced paroxysmal activity in rat hippocampal cell cultures”.

  1. Abstract sounds too general, not sharp enough. I would start by specifying what precise “pivotal role” AMPA receptors are proposed to play and by stating which particular mGluR agonists/antagonists were used. This is mainly to make the Abstract more attractive to potential readers browsing the data bases and, hopefully, increasing the impact, if the paper is published.

The abstract has been revised and the required details have been added. Please, see the revised manuscript.

Introduction

  1. Introduction is too long, almost review-like and, at the same time, leaves some important information out. It should emphasize the points which prompted the study. Let’s not forget that Glutamate, GABA, AMPA, kainate receptors (and more) in the mechanisms of epilepsy would be an old hat to many readers who may be looking for a fresh approach and new information.
  2. Line 45: Yes, indeed, there is strong evidence that even GABA itself can act as a maturation factor. See eg. Wolff et al.  Adv Exp Med Biol 181 215-239, 1984 (doi: 10.1007/978-1-4684-4868-9_17 .) and several more papers from the same laboratory in the 1980’s and 1990’s. One must wonder whether a GABA deficit or malfunction of GABA systems during development do not predispose to epilepsy. Maybe that the topic should be revisited.
  3. Line 68: “predominantly localize”, not the happiest formulation. Kainate receptors, to my knowledge, occur both pre- and postsynaptically; see Falcon-Moya R et al: Front. Mol. Neurosci., 22 June 2018 Sec. Brain Disease Mechanisms Volume 11 – 2018 https://doi.org/10.3389/fnmol.2018.00217 . The proposal that kainate receptors act on neurotransmitter release has been considered for a long time, too, but it is probably not their only role.
  4. Line 94: Introduction should summarize the state of the art in the field, formulate a hypothesis to be tested and state how it will be done. In other words, readers expect more of what “we intend to examine/test” rather than what “we have demonstrated”, the latter being better left for Conclusion. The findings belong to Results and their significance to Discussion. As it stands, we have covered nearly one hundred lines and, so far, we have no word on what exactly we want to do and what experimental model we shall be using, not to mention the experimental animal.

We thank the Reviewer for these remarks. We agree that this section was overloaded. Introduction has been rewritten and significantly shortened. Please, see the revised manuscript. This work was more focused on filling gaps in experimental data than on testing a specific conceptual hypothesis.

Other comments

  1. Line 54: “assumption is confirmed”, not the best form. Try – “this hypothesis has been tested” or “this proposal has been further buttressed” or simply “this is further supported by” see also L. 378

We thank the Reviewer for this remark. The sentences have been corrected.

  1. Lines 61/62 ”myriads.. ..mechanisms,” This sounds very clumsy, hard to understand. Try “..in a large number of neurons firing synchronously and involves many distinct mechanisms such as communications..”

We thank the Reviewer for this suggestion. The sentences have been rewritten.

  1. Lines 95/96: Historically, bicuculline is certainly the most important GABA antagonist and it was instrumental in identifying GABA as the main inhibitory neurotransmitter (Neurochem Int 35 269-280, 1999 but it has problems with solubility and stability as well as specificity (e.g. Breuker E, Johnston GAR, J Neurochem 25 903-904, 1975). Gabazine (SR95531) seems to be a more specific GABA antagonist (there should not be many inhibitory gabazine-sensitive glycine receptors in the current experimental model) and is much easier to handle. Have you tried it?

No, we have not used gabazine. We have chosen bicuculline since this antagonist has been used in numerous studies of epileptiform activity in vitro. In addition to bicuculline, we have also tested non-competitive GABA(A)R antagonist, picrotoxin. As with bicuculline, picrotoxin induced the [Ca2+]i oscillations in neurons, but in contrast to bicuculline, this effect attenuated but remained even after the triple wash out.

  1. Line 128: “the next”, better “the following”
  2. Line 166: “in dependence”, better “depending on”

Points 10-11. The corrections have been made.

  1. Line 307: “non-synaptic mechanisms” Gap junctions form "electrical synapses" so the communication by Gap Junctions is, to be physiologically precise, not "non-synaptic" (Pereda AE, Nature Reviews Neuroscience  15, 250–263, 2014).

We thank the Reviewer for this comment. This sentence has been corrected.

  1. Line 317: “..we have shown.. ..promising antiepileptic drugs, ..” It was actually shown years ago by others (reviewed, e.g., in Int Rev Neurobiol 119, 233-255, 2014, doi: 10.1016/B978-0-12-801022-8.00011-8) . So, what is really new in your conclusion?

The range of studies demonstrates that A1R agonists and agonists of some other Gi-coupled receptors can suppress the epileptiform activity, including bicuculline-induced. However, we focus on the ability of the agonists to prevent the generation of epileptic discharges. It is critically important for the prevention of epilepsy, which development occurs after traumatic brain injury. So, we demonstrate that only activation of A1Rs prevented the bicuculline-induced paroxysmal activity.

  1. Discussion: There is a potentially very important element missing in the evaluation of the data. Discussion aims mainly at receptors and completely disregards the presence of any neurotransmitter inactivation system. I mean the glutamate transporters such as GLT1 or GLAST, present in astrocytes (recent comprehensive reviews: Verchratsky & Niedergaard, Physiol Rev 98, 239-389, 2018; Andersen & Schousboe J Neurochem doi: 10.1111/jnc.15811, 2023). Yet astrocytes are obviously present in the cell culture and the neurotransmitter transport may play a very significant part in the formation and modification of the synaptic events (and in the mechanisms of epilepsies, Tanaka et al, Science 276 1699-1702, 1997). There is a more specific problem; in the cell culture, neurotransmitter may rapidly diffuse away from the site of action at the synapses which are exposed to a very large “extracellular space” i.e. the space above the monolayer culture filled with medium (thus creating a steep neurotransmitter concentration gradient above the “two-dimensional” cell culture). The role of transporters may appear, under such circumstances, limited; one could be tempted to posit (misleadingly) neurotransmitter release (at the expense of neurotransmitter re-uptake) as the most important determinant of the neurotransmitter concentrations at the synapse. In contrast, the extracellular space in vivo is much smaller, diffusion is hindered and not fast enough (Sykova & Nicholson Physiol Rev 88 1277-1340, 2008); the transport may then become much more important for shaping the synaptic activity.  This can be illustrated by somewhat conflicting conclusions drawn from two different types of experiments, both looking at the role of transporters, but using different models; Mennerick & Zorumski  J Neurosci 15 3178-3192, 1995, cell cultures) question the significance of transporters at the synapse while Diamond & Jahr J Neurophysiol 83 2835-2843, 2000, hippocampal slices preserving the three-dimensional structure of the tissue) ascribe the glutamate transporters a direct role in the synaptic function. Authors should, at the very least, remind the readers of a potential role of glutamate (and GABA) transporters in modifying the epilepsy-related mechanisms, at least in vivo or in alternative experimental models. This would not necessarily invalidate their present conclusions, just limit them to the current experimental model thus avoiding possible future misunderstandings and/or false claims.  

Many thanks for this remark. The necessary explanations have been added into Discussion.

  1. One other comment related to the above point: Kainate also inhibits glutamate transport (Johnston et al, J Neurochem 32, 121-127, 1979) and authors may wish to acknowledge it.

We thank the Reviewer for this remark. However, most of the recent studies demonstrate that high doses of kainic acid block glutamate release, whereas the correlation between the activity of KARs and glutamate uptake is unclear. Hence, the effects of the kainate on the glutamate uptake in brain slices can be explained only by the differences in affinity of kainate and glutamate for neuronal and glial glutamate transporters.

  1. It is not entirely clear to me why the glutamate release from astrocytes has to be discussed in such a detail, particularly when it was not directly investigated. I accept that the photolytic liberation of Ca2+ from its caging may have been linked to the release of glutamate from astrocytes in other studies (ref. 27 and 29) but it has not been looked at in the present experiments. In any case, how would you tell the difference between neuron-originating and astrocyte-originating glutamate? Additionally, glutamate could be released from astrocytes in a non-vesicular manner by reversed transport process (Szatkowski et al Nature 348 443-446, 1990). This sometimes happens in mammalian glia, too, particularly under the conditions of ischemia, or, generally, when the cells are metabolically compromised (could it happen during prolonged epileptic discharge?).

As known, astrocytic glutamate release is a process which is accompanied by the changes in intracellular Ca2+ concentration ([Ca2+]i). In the case of vesicular glutamate secretion, Ca2+ ions are directly involved in the release process. In turn, reverse flow of glutamate through the glutamate transporters or anion channels (Best-1) is accompanied by the changes in [Ca2+]i, but the gradients of other ions, such as Na+ or Cl-, are considered as the driving force of this process. However, it cannot be unambiguously concluded whether calcium ions are involved in the non-vesicular release of glutamate by astrocytes. Nevertheless, we did not observe any detectable fluctuations in [Ca2+]i in astrocytes after bicuculline application. Moreover, as we have previously shown (10.1002/glia.23763), tonic glutamate release by astrocytes does not occur in our neuron-glial cell cultures. We did not observe slow-inward currents (SICs) in neurons during the holding of membrane potential at -30 mV in the presence of tetrodotoxin (this protocol is used do discriminate the currents caused by astrocytic GABA and glutamate). Astrocytic glutamate release did not also occur, even in the presence of bicuculline. In turn, we observed bicuculline-sensitive slow-outward currents (SOCs) mediated by astrocytic GABA. Thus, we conclude that the contribution of astrocytic glutamate is negligible in our in vitro model of the paroxysmal activity. The necessary explanation has been added into Discussion.

  1. The finding that the cultured neurons display, individually and collectively, types of behaviour reminiscent of phenomena associated with epilepsies is, of course, encouraging. It would be good though, to mention (possibly in Conclusion or in a separate section on ”Limitations of the Study”) that cultures grown in Petri dish (and/or on coverslips) in the presence of a potential convulsant (penicillin) may not faithfully represent three-dimensional structure of brain (hippocampal) tissue (cf. # 14 above) and the present conclusion need to be further tested using additional experimental models. This might appear to devalue the outcome of the study somewhat, but it could pre-empt hostile criticism and make the final arguments (suitably modified) more credible.

We thank the Reviewer for this advice. We have added the necessary information into the Conclusion. Regarding the proconvulsant effect of penicillin, it should be noted that we did not observe spontaneous activity (synchronous oscillations of [Ca2+]i) in neurons without any exposures. These oscillations can be induced by Mg2+-free medium (DOI: 10.1016/j.eplepsyres.2019.106224), NH4Cl, or bicuculline (10.1002/glia.23763). Moreover, we have shown that neurons in our cultures undergo significant tonic inhibition (10.1002/glia.23763). Thus, we suppose that the used doses of penicillin do not induce epileptiform activity.

  1. MS would benefit from a list of abbreviations. Supposing I want to know the meaning of “HCN” and “SAG” (Fig. 4 and lines 181 and 192)?

We have defined all the abbreviations (excluding commercial names of some drugs) in the text.

Reviewer 3 Report

The research paper is interesting and well written, but I have a few comments:
- please explain the abbreviations used: AMPA, NMDA, cAMPA
- quite a few of the publications used are older, so it is worth checking for new reports on the subject in recent years (2022-2023)

Author Response

The research paper is interesting and well written, but I have a few comments:
- please explain the abbreviations used: AMPA, NMDA, cAMPA

All abbreviations have been defined. Please, see the revised version of the manuscript.
- quite a few of the publications used are older, so it is worth checking for new reports on the subject in recent years (2022-2023)

We agree with the Reviewer that one third of the cited works were published more than 10 years ago. Unfortunately, most of the recently published works on the issues raised in the article are review articles, accumulating mainly experimental studies carried out in 1990-2010. This is particularly true for articles on the role of kainate receptors. For this reason, we have cited only older primary sources.

Round 2

Reviewer 2 Report

I accept authors' changes and/or explanations. As to the abbreviations, I believe that a separate list would have been more courteous to potential readers.

I recommend careful re-reading and spell/grammar check (see e.g. line 58, revised version "extremely").

Author Response

  • The authors appreciate the Reviewer`s comment about the separate abbreviation list. However, according to IJMS "Instructions for Authors", the abbreviations should be defined when they appear first time in the text. As we understood, a separate list of abbreviations is not typical for the journal.
  • The text has been thoroughly revised. The version of the manuscript with marked changes has also been submitted.